# Offline Reinforcement Learning for Customizable Visual Navigation

**Dhruv Shah**[†]**, Arjun Bhorkar**[†]**, Hrishit Leen, Ilya Kostrikov, Nick Rhinehart, Sergey Levine**
UC Berkeley

**Abstract:** Robotic navigation often requires not only reaching a distant goal, but also satisfying intermediate user preferences on the path, such as obeying the rules of the road or preferring some surfaces over others. Our goal in this paper is to devise a robotic navigation system that can utilize previously collect data to learn navigational strategies that are responsive to user-specified utility functions, such as preferring specific surfaces or staying in sunlight (e.g., to maintain solar power). To this end, we show how offline reinforcement learning can be used to learn reward-specific value functions for long-horizon navigation that can then be composed with planning methods to reach distant goals, while still remaining responsive to user-specified navigational preferences. This approach can utilize large amounts of previously collected data, which is relabeled with the task reward. This makes it possible to incorporate diverse data sources and enable effective generalization in the real world, without any simulation, task-specific data collection, or demonstrations. We evaluate our system, ReViND, using a large navigational dataset from prior work, without any data collection specifically for the reward functions that we test. We demonstrate that our system can control a real-world ground robot to navigate to distant goals using only offline training from this dataset, and exhibit behaviors that qualitatively differ based on the user-specified reward function.

**Keywords:** offline reinforcement learning, visual navigation

## 1 Introduction

Robotic navigation approaches aim to enable robots to navigate to user-specified goals in known and unknown environments. The *geometric* approach to this problem involves using a geometric map of the environment to plan a collision-free path towards the goal. The *learning-based* approach to this problem involves the robot learning to take actions by associating new inputs with prior navigational experience, typically through imitation learning (IL) or reinforcement learning (RL). In many practical navigational scenarios, the goal is not merely to *reach* a particular destination without collision, but to do so while maximizing some desired *utility measure*, which could include obeying the rules of the road, staying in a bike lane or off the lawn, maintaining safety, or even more esoteric goals such as remaining in direct sunlight for a solar-powered vehicle. In these cases, neither IL nor geometric navigation alone would suffice without strong assumptions on the nature of the task or task-specific near-optimal demonstrations, which may be difficult to obtain. RL can, in principle, address such challenges, but prior work on applying RL to robotic navigation relies on infeasible amounts of online data collection, or requires high-fidelity simulators for simulation to real world transfer. Is there a practical RL paradigm than can solve this challenge directly from real-world data?

---

[†] These authors contributed equally.

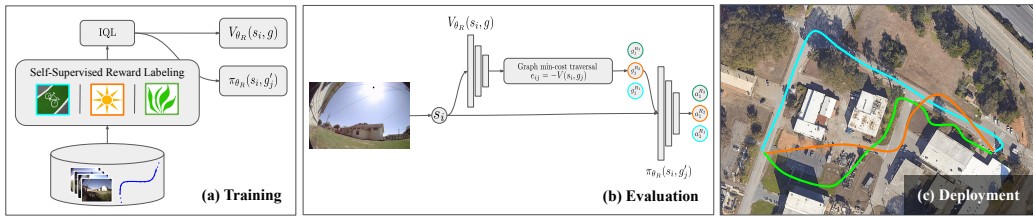

**Figure 1: Customizable navigation with ReViND:** (a) During training, we use an offline dataset of trajectories in conjunction with reward labels, specifying a target utility, to learn a Q-function. (b) During deployment, we use the values from this Q-function to generate a topological graph over observations, and plan through it. This plan incorporates the desired utility objective and can demonstrate varied behaviors (c) such as goal-reaching while driving in the sun, driving on grass, or more abstract objectives like following a bike lane.

RL from offline datasets [1] can address this challenge by learning policies from a dataset of previously collected trajectories. Importantly, given a previously collected diverse dataset of navigational trajectories, it is possible to relabel that dataset post-hoc with any reward function we want. Thus, a user could gather a large dataset from a variety of prior sources (e.g., experiments in similar environments, or even from the web), label it with a custom reward function, train their own policy, and then deploy it, all *without ever having to collect their own data*. Since this approach can leverage large datasets, it may lead to significantly better generalization [2] than methods that require much more tightly curated data, such as imitation learning methods. How can we design a system to learn control policies from large datasets that can be immediately deployed onto a mobile robot?

In this paper, we describe a robotic learning system that performs visual navigation to distant goals (e.g. 100s of meters away) while also incorporating user-specified reward objectives. Our system consists of two parts: (i) an offline Q-learning algorithm [3] that can incorporate the desired preferences in the learned Q-function and trains a policy operating directly on raw visual observations, and (ii) a topological representation of the environment, where nodes are represented by the raw visual observations and the connectivity between them is described by the learned value function (see system overview in Fig. 1). While the Q-function alone may only be sufficient to learn accurate navigational startegies over short horizons [4], combining it with a topological graph allows scaling to large environments by searching for a sequence of nodes to the goal that maximizes the desired objective at a coarse-level. The policy derived from the Q-function is subsequently used to navigate between the nodes, while also maximizing the specified objective at the lower-level.

The primary contribution of this work is ReViND, a robotic system for **Re**inforcement learning for **Vi**sual **N**avigation with prior **D**ata, that can utilize previously collected data in real-world environments, adopt behavior that is well-suited to user-specified reward functions (e.g., staying on the grass, staying in sunlight), and still be able to reach distant goals by combining planning and learning. We demonstrate that ReViND can readily incorporate high-level rewards, such as preferences for particular surfaces or staying in sunlight, and can perform goal-reaching in complex environments over hundreds of meters. ReViND is pre-trained on 30 hours of publicly available navigation data [5] and is deployed in a visually similar target environment *without* any on-policy data collection or fine-tuning. To the best of our knowledge, our results are the first demonstration of offline RL for navigation utilizing only previously available public datasets. Our experiments show that ReViND demonstrates markedly different goal-reaching behaviors by tweaking the reward objectives, while outperforming policies trained with imitation learning and model-free reinforcement learning.

## 2  Related Work

Robotic navigation has been studied from perspective of mapping, planning, imitation learning, and reinforcement learning. Classical navigation methods first acquire a geometric map, and then use this map to plan collision-free paths [6]. The map can be built up incrementally, including with intelligent exploration strategies that maximize information gain or otherwise optimize for faster map acquisition [7–18]. Such methods often aim to map out the entire environment first and then

execute specific navigational tasks, though it is also possible to perform target-driven exploration, where the map is built in the course of navigating to a specific goal [19, 20]. However, all such methods are fundamentally geometric: the task is defined as reaching some destination, rather than in terms of semantic reward functions that we consider in this work.

Similarly, many learning-based approaches to navigation also focus on largely geometric tasks, such as the "PointGoal" task [21], though they often utilize reinforcement learning methods that in principle can accommodate any reward function. The more severe issue with such methods is that RL algorithms can require a large amount of online experience (e.g., millions or even billions of trials) [22]. A method that requires a million 1-minute episodes would take more than 1.5 years of nonstop real-world collection, making it poorly suited for learning from scratch directly in the real world. Therefore, such methods typically require simulation, transfer, and other additional components [23–25]. An alternative for encoding specific user preferences into a learning-based method is to employ imitation learning [26]. While imitation learning can enable a user to define their desired behavior through the demonstrations, such demonstrations are time-consuming to gather, and must be recollected for each new reward function. In contrast, our offline RL method utilizes previously collected datasets, which we show is practical for real-world robots, and relabels the same dataset with different reward functions, which means no reward-specific data collection is needed.

Prior offline RL work has proposed a number of algorithms that can utilize previously collected data [1, 3, 27, 27–31]. Our goal is not to develop a new offline RL algorithm, but rather to explore their application to robotic navigation tasks. Most prior robotics applications of such methods include multi-task learning for manipulation [32–36]. Unlike these works, our focus is specifically on how a single dataset can be reused to enable long-horizon navigation with different user preferences. To that end, we combine our approach with graph-based search to reach distant goals, which we show significantly improves over direct use of the learned policy, and utilize the same exact data to optimize different reward functions. While prior work has also explored the use of offline data with varying reward functions [37, 38], we address significantly longer horizon tasks by incorporating model-free RL and graph search.

Our use of graph search in combination with RL parallels prior work that integrates graph search and planning into supervised skill learning methods [26, 39] and goal-conditioned reinforcement learning [4, 40, 41]. However, our method differs from these works in two ways. First, while these prior works use the value function to estimate the temporal distance between pairs of nodes in the graph, we specifically exploring using a variety of objectives. More importantly, we use offline RL, whereas prior work uses either supervised regression for distances, or online RL. Our goal is not to develop a new offline RL algorithm, but to explore their application to robotic navigation tasks by building a learning-based system that can use one of these algorithms for planning. To our knowledge, our work is the first to combine topological graphs with RL for arbitrary reward functions, and the first to combine them with offline RL.

## 3 Offline Reinforcement Learning for Long-Horizon Robotic Navigation

Our system combines offline learning of reward-specific value functions with topological planning over a graph constructed from prior experience in a given environment, so as to enable a robot to navigate to distant goals while maximizing user-specified rewards. The learned value function is used not only to supervise a local policy that chooses reward-maximizing actions, but also as a way to evaluate edge costs on a graph constructed from past experience. The graph is then used to plan a path, and the policy is used to execute the action to reach the first subgoal on that path. Structurally, this resembles SoRB [4], but with two critical changes: the use of offline RL to learn value functions from data, and the ability to handle various reward functions rather than just finding minimum length paths.

## 3.1 Problem Statement and Assumptions

The robot's task is defined in the context of a goal-conditioned MDP, with state observations $s \in \mathcal{S}$, actions $a \in \mathcal{A}$, and goals $g \in \mathcal{G}$. The robot receives a reward $r(s_t)$ at each time step $t$, which depends on the degree to which it is satisfying user preferences (e.g., staying on the graph), with an additional bonus for reaching the goal. The objective can be expressed as maximizing the total reward of the robot's executed path, since the reward accounts for both the desired utility and goal reaching. We discuss specific rewards in Section 4. The state observations consist of RGB images from the robot's forward-facing camera and a 2D GPS coordinate, the actions are 2D steering and throttle commands, the goal is a 2D GPS coordinate expressed in the robot's frame of reference. In this setting, reinforcement learning methods will learn policies of the form $\pi(a_t|s_t, g_t)$, though our approach will not command the final task goal $g_t$ directly, but instead will use a planning method to determine intermediate subgoals, which in practice makes it significantly easier to reach distant goals. To enable this sort of planning, we make an additional assumption that parallels prior work on combining RL with graph search [4, 26]: we assume that the robot has access to prior experience from the current environment that it can use to build a topological graph that describes its connectivity. Intuitively, this corresponds to a kind of "mental map" that describes which landmarks are reachable from which other landmarks. Importantly, we do *not* assume that this graph is manually constructed or provided: the algorithm constructs the graph automatically using an uncurated set of observations recorded from prior drive-throughs of the environment. In our experiment, these traversals are done via teleoperation, though they could also be performed via autonomous exploration, and our method could be extended to handle unseen environments by integrating the exploration procedures discussed in prior work [5].

## 3.2 Reinforcement Learning from Offline Data

Offline RL algorithms learn policies from static datasets. In our implementation we use implicit Q-learning (IQL) [3], though our approach is compatible with any value-based offline RL algorithm. We summarize offline RL in general and IQL specifically in this section. Given a dataset $\mathcal{D} = \{(s_i, a_i, r_i, s_i') \mid i = 1 \ldots N\}$, the goal of offline RL is to learn a policy that optimizes the sum of discounted future rewards without any additional interactions with the environment. IQL involves fitting two neural networks, $Q_\theta$ and $V_\psi$, where $Q_\theta(s, a, g)$ approximates the $Q$-function of an *implicit* policy that maximizes the previous Q-function, and $V_\psi$ represents the corresponding value function. The Q-function is updated by minimizing squared error against the next time step value function, with the objective

$$L(\theta) = \mathbb{E}_{(s,a,s') \sim \mathcal{D}, g \sim p(g|s)}[(\gamma V_\psi(s', g) + r(s, a, g) - Q_\theta(s, a, g))^2],$$

where $p(g|s)$ is a goal distribution, which we will discuss later. The value function $V_\psi(s, g)$ should be trained to correspond to $Q_\phi(s, a, g)$ for the optimal action $a$ that maximizes the value at $s$, but directly computing $\max_a Q_\phi(s, a, g)$ is likely to select an "adversarial" out-of-distribution action that leads to erroneously large values, since the static dataset does not permit $Q_\phi(s, a, g)$ to be trained on all possible actions [1, 3, 30]. Therefore, IQL employs an *implicit* expectile update, with a loss function given by

$$L(\psi) = \mathbb{E}_{(s,a) \sim \mathcal{D}, g \sim p(g|s)}[L_2^\tau(Q_\theta(s, a, g) - V_\psi(s', g))],$$

where $L_2^\tau(u) = |\tau - \mathbb{1}(u < 0)|u^2$. This can be shown to approximate the maximum over *in-distribution* actions [3], but does not require ever querying out-of-sample actions during training. To instantiate this method, it remains only to select $p(g|s)$.

**Goal relabeling.** The IQL algorithm is not goal-conditioned [3], and the dataset was not collected with a goal-reaching policy, so the goals must be selected post-hoc with some sort of *relabeling strategy*. While a variety of relabeling strategies have been proposed in prior work [4, 36, 42–44], we follow prior work on offline RL for goal-reaching [36] and simply set the goal to states that are observed in the dataset in the same trajectory at time steps subsequent to a given sample $s_i$. In our implementation, we select this time step at random between 10 and 70 time steps after $s_i$ (the total

trajectory lengths are typically around 80 steps). Algorithm 1 outlines pseudocode for training the Q-function with IQL.

**Long-horizon control.** Instead of directly using the policy learned with IQL, in this paper we use the IQL value function to obtain edge costs for a graph used for topological planing. The standard IQL method directly extracts a reactive policy from the Q-function. However, we found that in the real world, this approach was unable to reach goals farther than 20m, or 80 time steps. A deeper analysis of the system revealed that, while the policy and values learned by the IQL agent are valid over shorter horizons, they degrade rapidly as the horizon increases. This is not surprising, because like all value-based methods, IQL assumes that $s$ represents a Markovian state. But this assumption becomes increasingly violated for long-horizon tasks with first-person images: while goals that are within line of sight of the robot are relatively simple, goals that require navigating around obstacles tend to fail if using the policy directly. In the next subsection, we will discuss how we can use a topological graph as a sort of "nonparametric memory" of the environment to alleviate this challenge, enabling our method to reach distant goals.

### 3.3 Long-Horizon Reward Maximization with a Topological Graph

To enable long-horizon navigation, we combine the value function learned via offline RL with a topological graph built from prior observations in a given environment. As discussed previously, we assume that the robot has a limited amount of prior experience in the test environment that can be used to build a "mental map," corresponding to a graph where nodes are observations and edges represent the cumulative reward the robot will accumulate as it travels from one node to another. Note that this graph is topological rather than geometric: the nodes are image observations, and the connectivity is determined by the learned value function. We do not use the data from the test environment to finetune the value functions, only to construct the graph.

The graph $\mathcal{G}$ is constructed in the same way as prior work on graph-based navigation (see Shah et al. [26] for the closest prior method): each state observation $s_i$ in the test environment corresponds to a node $n_i$, and each edge $e_{ij}$ receives a cost corresponding to $C(e_{ij}) = -V_\psi(s_i, s_j)$.[1] We further filter these edges based on the GPS coordinates of the nodes to eliminate *wormholes* arising due to optimistic value estimates. For more details on how the graph is constructed, please see Appendix B. Given an overall task goal, we add it to the graph as an additional node $n_G$, along with a node representing the robot's current state, and then use Dijkstra's algorithm to compute the shortest path with these edge costs. We then use the policy learned via offline RL to navigate to the first node along this path. Algorithm 2 outlines pseudocode for this procedure.

In our implementation, we use goal-conditioned reward functions of this form:

$$R(s_t, a_t, g) = \begin{cases} -k_t(s_t) & \forall s_t \neq g \\ 0 & \text{otherwise.} \end{cases} \tag{1}$$

where $k_t(s_t) > 0$ is always positive to ensure that the planner actually reaches the goal.

**Proposition 3.1** *If we recover the optimal value function $V^*(s, s')$ for short-horizon goals $s'$ (relative to $s$), and $\mathcal{G} = \mathcal{S}$ (all states exist in the graph), and the MDP is deterministic with $\gamma = 1$, then finding the minimum-cost path in the graph $\mathcal{G}$ with edge-weights $-V^*(s, s')$ recovers the optimal path, that is, a policy $\pi$ that maximizes $V^*(s, g)$.*

*Proof (sketch)*: The Bellman equation can be used to write the cost of the minimal-cost path in the graph with edge-weights $-V(s, s')$: $J^*(s, g) = \min_{s'}[-V(s, s') + J^*(s', g)] = -\max_{s'}[V(s, s') - J^*(s', g)]$. We can further expand $V(s, s')$ into a sum of rewards induced by the policy $\pi$ and then rearrange the terms to obtain a similar optimality equation for $V^*$ that demonstrates that $J^*(s, g) = -V^*(s, g)$.

---

[1]In our actual implementation, goals are defined only in terms of GPS coordinates, rather than the full image, so technically the second argument is only the GPS coordinate of $s_j$, which we found to be sufficient, though extending the method to use the full image is a simple modification

While the above proposition is in some sense an obvious statement (and derived under strong assumptions), it provides some degree of confidence that our proposed method is *correct* and *consistent*, in the sense that in the limit of unlimited data it will eventually obtain the optimal solution.

---

**Algorithm 1** Training ReViND

1: Initialize parameters $\psi, \theta, \hat{\theta}, \phi$.
2: **for** each gradient step **do**
3:     Sample a mini-batch $\{(s_i, a_i, r_i, s_i')\}$
4:     **for** each sample **do**
5:         $T \leftarrow T \in D \mid s_i \in T$
6:         $g_i \leftarrow \text{SampleGoal}(T, s_i)$
7:         $s_i, s_i' \leftarrow \text{Relabel}(s_i, s_i', g_i)$
8:         $r_i \leftarrow \text{Reward}(s_i, g_i)$
9:     $\psi \leftarrow \psi - \lambda_V \nabla_\psi L_V(\psi)$
10:    $\theta \leftarrow \theta - \lambda_Q \nabla_\theta L_Q(\theta)$
11:    $\hat{\theta} \leftarrow (1 - \alpha)\hat{\theta} + \alpha\theta$

---

**Algorithm 2** Deploying ReViND

1: **Inputs**: current observation obs := $\{\text{img}, x\}$, set of past observations $\mathcal{N} := \{n_1, \ldots, n_m\}$, IQL agent $\{Q, V, \pi\}$, goal node $n_G \in \mathcal{N}$
2: $\mathcal{G} \leftarrow \text{ConstructGraph}(\mathcal{N}, V)$
3: **while** not IsClose(obs, $n_G$) **do**
4:     UpdateGraph(obs)
5:     $w_1, \ldots, w_k \leftarrow \text{DijkstraSearch}(\text{obs}, n_G)$
6:     **for** $t = 1, \ldots, H$ **do**
7:         goal vector = GetRelative($x, w_1$)
8:         RunPolicy(img, goal)    ▷ runs on robot
9:         obs ← next observation

---

## 4 System Evaluation

We now describe our system and experiments that we use to evaluate ReViND in real-world environments with a variety of utility functions. Our experiments evaluate ReViND's ability to incorporate diverse objectives and learn customizable behavior for long-horizon navigation, and compare it alternative methods for learning navigational skills from offline datasets.

### 4.1 Mobile Robot Platform

We implement ReViND on a Clearpath Jackal UGV platform (see Fig. 1). The sensor suite consists of a 6-DoF IMU, a GPS unit for approximate localization, and wheel encoders to estimate local odometry. The robot observes the environment using a forward-facing $170°$ field-of-view RGB camera. Compute is provided by an NVIDIA Jetson TX2 computer, with the offline RL controller running on-board. Our method uses only the monocular RGB images from the on-board camera and unfiltered GPS measurements.

### 4.2 Offline Trajectory Dataset and Self-Supervised Labeling

The ability to utilize offline datasets enables ReViND to learn navigation behavior directly from existing datasets — which may be expert teleoperated or collected via an autonomous exploration policy — without collecting *any* new data. While it may be extremely challenging to get generalization capabilities that work for *all* scenarios, we demonstrate that ReViND can learn behaviors from a small offline dataset and generalize to a variety of previously unseen, *visually similar* environments including grasslands, forests and suburban neighborhoods. Expanding this training dataset to include more diverse scenes can help extend these results to alternate applications (e.g. indoors). To emphasize this, we train ReViND using 30 hours of publicly available robot trajectories collected using a randomized data collection procedure in an office park [5]. To utilize this data with our method, we relabel it with several different reward labels corresponding to three behaviors: simple shortest-path goal-reaching, driving in the sun (to emulate a solar-powered vehicle that needs sunlight), and driving on grass. We automatically generate labels for "sunny" and "grassy" observations by simple image processing operations in the HSV color space. We implement these rewards via additive bonus to the negative rewards which corresponds to reducing the penalty for traversing these areas. For more details, see Appendix A. Note that the relabeling process is fully autonomous using these rewards, although learning more intricate behaviors may require defining special rewards or manual labeling. As discussed in Sec. 3.2, we use IQL to learn the value functions and policies for each task.

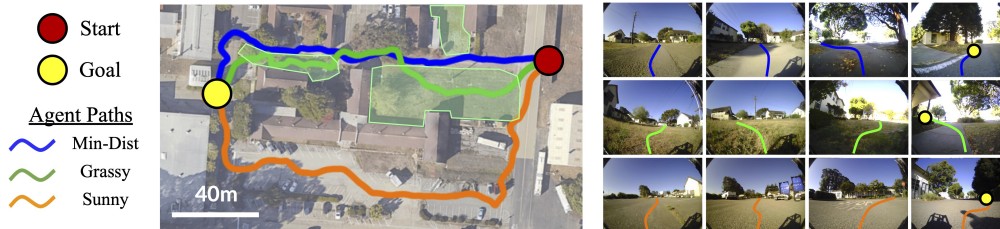

**Figure 2: Comparison of policies for different reward functions learned by ReViND.** Left: an overhead map (not available to the method), with grassy areas indicated with green shading. Note that the policy for the "sunny" reward chooses a significantly different path through a concrete parking lot without tree cover, while the policy for the "grassy" reward takes frequent detours to drive on lawns. Right: first person images during each traversal, with the chosen path indicated with colored lines.

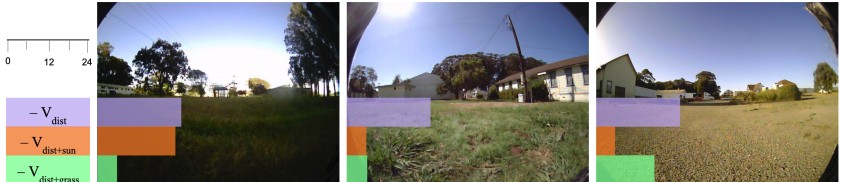

**Figure 3:** The learned values for different utilities, shown here as horizontal bar plots, can incorporate diverse objectives, and are used downstream by ReViND to infer edge connectivity in the topological graph. E.g. a solar robot would perceive goals in a sunny environment (right) as favorable due to lower distance estimates, and prefer a route that avoids shadows.

Figure 3 illustrates the values learned by the Q-function for reaching goals that are ~10m away. The value estimates corresponding to the three different utility functions estimate markedly different traversability costs. For example, $V_{dist+sun}$ predicts a lower cost (i.e., a higher value) for sunny observations, indicating that this value function prefers such paths.

## 4.3 Learning Varied Behaviors with ReViND

We now evaluate our method both in terms of its ability to tailor the navigational strategy to the provided reward, and in terms of how it compares to prior approaches and baselines. We test ReViND in five suburban environments for a large number of goal-reaching tasks (see Appendix H). While these environments are visually similar to the offline training data, they exhibit dynamic elements such as humans, moving automobiles, and changes in the appearance of grass and trees across the seasons. In each evaluation environment, we first construct a topological graph by manually driving the robot and collecting visual and GPS observations. The nodes of this graph are obtained by sub-sampling these observations, such that they are 10–30m apart, and the edge connectivity is determined by the corresponding value estimates. Note that the Q-function is *not updated* with this data, it is only used to build the graph. Once the graph is constructed, the robot is tasked with reaching a goal location, where it follows Alg. 2 to search for a path through the graph, and then executes it via the learned policy. Fig. 2 shows the paths taken by policies corresponding to the different reward functions for a specific start-goal pair. The overhead image is not available to ReViND and is only provided for illustration.

Our results show that utilizing value functions for different rewards from ReViND leads to significantly different paths through the environment. For example, the "sunny" reward function causes a large detour through a parking lot without tree cover, while the "grassy" reward causes frequent detours to drive on lawns. All of the policies successfully avoid obstacles and collisions and successfully reach the goal. In Table 1 we provide a quantitative summary of the behavior of the method for each reward function, showing success weighted by path length (SPL, which corresponds to an optimality measure that awards higher scores to successful runs with the shortest route length), the average value of the grass reward, and the average value of sun reward for trials corresponding to each reward function (note that these rewards are normalized to maximum of 1). As expected, we see that the values of these metrics strongly covary with the commanded reward.

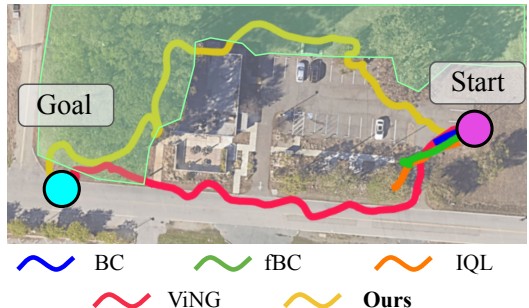

| | BC | | fBC | | IQL |
| ViNG | | | **Ours** | | |

**Figure 4:** Qualitatively, only ReViND reaches the goal while prioritizing grassy terrain (shaded green).

| Method | Uses Graph? | Easy (<50m) | | Medium (50–150m) | | Hard (150–500m) | |
| | | Success | $\mathbb{E}\mathbb{1}_{\text{grass}}$ | Success | $\mathbb{E}\mathbb{1}_{\text{grass}}$ | Success | $\mathbb{E}\mathbb{1}_{\text{grass}}$ |
|---|---|---|---|---|---|---|---|
| Behavior Cloning | No | 1/5 | 0.08 | 0/5 | 0.04 | 0/5 | 0.12 |
| Filtered BC | No | 3/5 | 0.29 | 0/5 | 0.08 | 0/5 | 0.12 |
| IQL [3] | No | 3/5 | 0.37 | 1/5 | 0.29 | 0/5 | 0.16 |
| ViNG [26] | Yes | **5/5** | 0.07 | **4/5** | 0.09 | 3/5 | 0.14 |
| Filtered BC + Graph | Yes | **5/5** | 0.24 | **4/5** | 0.15 | 3/5 | 0.19 |
| **ReViND (Ours)** | Yes | **5/5** | **0.47** | **4/5** | **0.84** | **4/5** | **0.78** |

**Table 2:** Success rates and utility maximization for the task of navigation un grassy regions ($R_{\text{grass}}$).

Next, we compare ReViND to four baselines, each trained on the same offline dataset. These approaches represent natural points of comparison for our method, and include prior imitation learning and RL methods, as well as a prior graph-based method that does not use RL. Since our approach is (to our knowledge) the first to combine RL with arbitrary

| Agent Utility | SPL | $\mathbb{E}R_{\text{grass}}$ | $\mathbb{E}R_{\text{sun}}$ |
|---|---|---|---|
| $R_{\text{dist}}$ | **0.87** | 0.16 | 0.61 |
| $R_{\text{dist}} + R_{\text{grass}}$ | 0.84 | **0.86** | 0.39 |
| $R_{\text{dist}} + R_{\text{sun}}$ | 0.64 | 0.05 | **0.68** |

**Table 1:** ReViND learns custom behaviors that maximize the desired utility.

rewards and topological graph search, no prior approach supports both graphs and arbitrary rewards. All methods have access to egocentric images and GPS, and command future waypoints to the robot.

**Behavioral Cloning (BC):** A goal-conditioned imitation policy that maps images and goals to control actions [45]. *This baseline does not incorporate reward information.*

**Filtered BC (fBC):** A similar goal-conditioned BC policy that incorporates reward information by filtering the training data, picking only trajectories with the top 50% aggregate rewards [46].

**ViNG:** A graph-based navigation system that combines a goal-conditioned BC policy and distance function with a topological graph [26]. *This baseline does not incorporate reward information.*

**IQL:** A baseline that uses only the learned Q-function, without a topological graph [3].

| Method | Easy (<50m) | | Medium (50–150m) | | Hard (150–500m) | |
| | Success | $\mathbb{E}\mathbb{1}_{\text{sun}}$ | Success | $\mathbb{E}\mathbb{1}_{\text{sun}}$ | Success | $\mathbb{E}\mathbb{1}_{\text{sun}}$ |
|---|---|---|---|---|---|---|
| Behavior Cloning | 1/5 | 0.58 | 0/5 | 0.32 | 0/5 | 0.29 |
| Filtered BC | 3/5 | 0.51 | 0/5 | 0.31 | 0/5 | 0.32 |
| IQL [3] | 3/5 | 0.54 | 2/5 | 0.42 | 0/5 | 0.34 |
| ViNG [26] | **5/5** | **0.63** | **4/5** | 0.58 | 3/5 | 0.63 |
| **ReViND (Ours)** | **5/5** | 0.61 | 3/5 | **0.75** | **4/5** | **0.74** |

**Table 3:** Success rates and utility maximization for the task of navigation in sunny regions ($R_{\text{sun}}$).

Fig. 4 shows the qualitative behavior exhibited by the different systems for maximizing the "grassy" reward function. IQL and Filtered BC can incorporate the reward function into the policy, but since they rely entirely on a reactive policy for navigation, they are unable to determine how to navigate toward the goal, and exhibit meainingless bee-lining behavior. Using a graph search to find a minimum distance path, ViNG can reach the goal, but does not satisfy the reward function. Only ReViND is successful in navigating to the goal while taking a short detour that maximizes the desired objective, demonstrating affinity to grassy terrains.

We provide a quantitative evaluation of these methods in Tables 2/3, showing the average distance traveled by each method over all test trials prior to disengagement (which occurs when there is high risk of collision), as well as the average value of the utilities. We see that non-RL methods are unable to take into account the task reward, and simply aim to reach the task goal, which leads to suboptimal utility. We can take reward into account either using RL, or by filtering BC to imitate only the high-reward trajectories. In easier environments, we see that both fBC and IQL can learn reward-maximizing behavior. However, both the RL and BC flat policies suffer sharp drops in performance as the distance to goal increases. The addition of a graph greatly helps improve performance. Here again, we notice that offline RL (ReViND), which uses Q-learning to optimize the reward, consistently outperforms filtering-based approaches (fBC-graph) – this confirms that taking the reward into account is important for respecting the user's preferences (which is unsurprising), but also that offline RL is more effective at this than simple filtering.

The biggest failure mode for current offline RL and IL methods in our task is their inability to reach distant goals. BC, fBC and IQL consistently fail to reach goals beyond 15-20m away, due to challenges in learning a useful policy from offline data — these *flat* baseline policies often demonstrate *bee-lining* behavior, driving straight to the goal, which often leads to collisions.

## 5  Discussion

We presented ReViND, a robotic navigation system that uses offline reinforcement learning in combination with graph search to reach faraway goals while optimizing arbitrary user-specified reward functions. We showed that ReViND can be trained on a navigational dataset collected in prior work and, when this dataset is relabeled with a variety of reward functions, ReViND can exhibit distinct behavior, preferring certain terrains or environment conditions during the traversal. Our experiments show that ReViND can generalize to novel, visually simlar environments, and is responsive to the specified reward function, and significantly outperforms prior methods that either do not utilize graph search and rely entirely on reactive policies, or else utilize graphs without RL and therefore do not support arbitrary rewards. We hope that our work will provide a step toward robotic learning methods that routinely reuse previously collected data, while still accomplishing new tasks and maximizing new user-specified reward functions. Such methods can exhibit effective generalization in the real world through their ability to incorporate large and diverse previously collected datasets, while still flexibly solving new tasks in a variety of visually similar environments, as long as the specified reward functions are valid in novel environments.

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

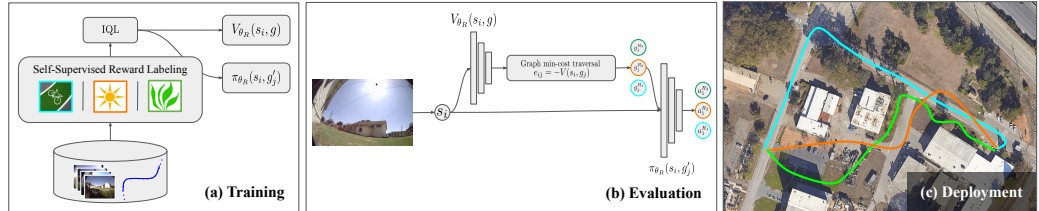

**Figure 1: Customizable navigation with ReViND:** (a) During training, we use an offline dataset of trajectories in conjunction with reward labels, specifying a target utility, to learn a Q-function. (b) During deployment, we use the values from this Q-function to generate a topological graph over observations, and plan through it. This plan incorporates the desired utility objective and can demonstrate varied behaviors (c) such as goal-reaching while driving in the sun, driving on grass, or more abstract objectives like following a bike lane.

# Part I

# Appendix

## Table of Contents

## Erratum

We noticed that Figure 1 in the submitted draft is extremely low resolution, likely due to a PDF compilation error. Please see a higher resolution copy above (reproduced without changes).

## A   Reward Labeling

For the base task of goal-reaching, we use a simple reward scheme with a *survival penalty* that incentivizes the robot to take the shortest path to the goal:

$$R_{\text{dist}}(s_t, a_t, g) = \begin{cases} -1 & \forall s_t \neq g \\ 0 & \text{otherwise.} \end{cases} \tag{2}$$

For more complex utilities, such as incentivizing driving in the sun (e.g., for a solar robot), we discount the survival penalty by a factor of 4.

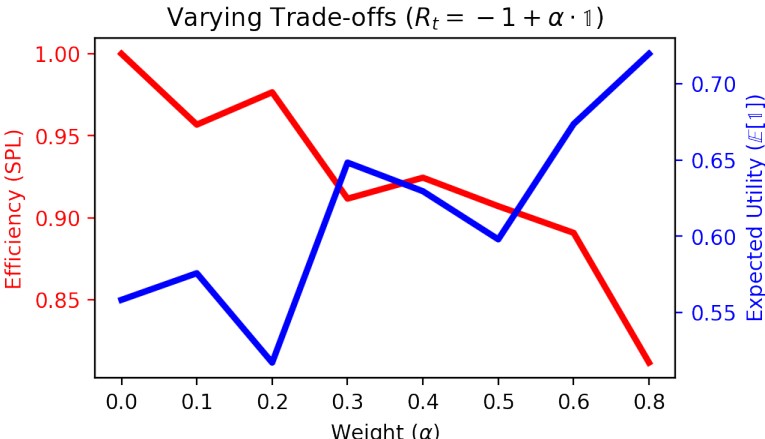

**Figure 5:** ReViND can support a wide range of reward functions and performs as expected for varying levels of trade-offs between the goal-reaching and utility maximization objectives.

$$R_{\text{grass}}(s_t, a_t, g) = \begin{cases} -1 + 0.75 * \mathbb{1}_{\text{grass}}\{s_t\} & \forall s_t \neq g \\ 0 & \text{otherwise.} \end{cases} \tag{3}$$

$$R_{\text{sun}}(s_t, a_t, g) = \begin{cases} -1 + 0.75 * \mathbb{1}_{\text{sun}}\{s_t\} & \forall s_t \neq g \\ 0 & \text{otherwise.} \end{cases} \tag{4}$$

An interesting implication of the above reward scheme is to view the negative penalty as a proxy for the amount of work a robot needs to do — a solar robot may use 1 unit of energy per time step to navigate in an environment, but it may also create 0.75 units of energy by exposing itself to the sun, effectively discounting the navigation cost in sunny regions. This reward scheme trades-off the choice of the shortest path to the goal with maximizing the user-specified utility function.

For our experiments, we use the robot's egocentric visual observations to automatically label these rewards. To determine whether the current state is in a sunny/grassy area, we process the front view image of the Jackal by thresholding in the HSV colorspace. We process the bottom center crop of the image by thresholding it, and declare event $\mathbb{1}_{\text{sun}}$ or $\mathbb{1}_{\text{sun}}$ if a majority of the pixels satisfy the thresholds.

We note that while the above choice of reward function may seem arbitrary, the overall utility function (or the "relative weight" between the two objectives) would be application-dependent. For instance, a solar-powered robot may be able to recoup 20% of its navigation energy when driving in the sun, and its effective reward could be $(-1 + 0.2 * \mathbb{1}_{\text{sun}})$. We ran experiments to test ReViND's sensitivity to this trade-off and found that it performs expectedly for a wide range of reward functions (see Figure 5). Practically, this would be a hyperparameter set empirically by the user based on the desired level of trade-off between the goal-reaching and utility maximization objectives.

## B  Building the Topological Graph

As discussed in Section 3.3, we combine the value function learned via offline RL with a topological graph of the environment. This section outlines the finer details regarding how this graph is constructed. We use a combination of learned value function (from Q-learning), spatial proximity (from GPS), and temporal proximity (during data collection), to deduce edge connectivity. If the corresponding timestamps of two nodes are close ($< 2s$), suggesting that they were captured in quick succession, then the corresponding nodes are connected — adding edges that were physically traversed. If the distance estimates (or, negative value) between two nodes are small, suggesting that

they are *close*, then the corresponding nodes are also connected — adding edges between distant nodes along the same route, and giving us a mechanism to connect nodes that were collected in different trajectories or at different times of day but correspond to the nearby locations. To avoid cases of underestimated distances by the model due to aliased observations, e.g. green open fields or a white wall, we filter out prospective edges that are significantly further away as per their GPS estimates — thus, if two nodes are nearby as per their GPS, e.g. nodes on different sides of a wall, they may not be disconnected if the values do not estimate a small distance; but two similar looking nodes 100s of meters away, that may be facing a white wall, may have a small distance estimate but are not added to the graph in order to avoid *wormholes*. Algorithm 0 summarizes this process — the timestamp threshold $\epsilon$ is 1 second, the learned distance threshold $\tau$ is 50 time steps (corresponding to $\sim 12$ meters), and the spatial threshold $\eta$ is 100 meters.

---

**Algorithm 3** Graph Building

---

1: **function** GETEDGE$(i, j)$
2:     **Input**: Nodes $n_i, n_j \in \mathcal{G}$; value function $V_\psi$; hyperparameters $\{\tau, \epsilon, \eta\}$
3:     **Output**: Boolean $e_{ij}$ corresponding to the existence of edge in $\mathcal{G}$, and its weight
4:     goal = GetRelative$(n_i, n_j)$                    ▷ using GPS and compass
5:     $D_{ij} = -V_\psi(n_i, \text{goal})$                   ▷ learned distance estimate
6:     $T_{ij} = |n_i[\text{'timestamp'}] - n_j[\text{'timestamp'}]|$        ▷ timestamp distance
7:     $X_{ij} = \|n_i[\text{'GPS'}] - n_j[\text{'GPS'}])\|$           ▷ spatial distance
8:     **if** ( $T_{ij} < \epsilon$) **then** return $\{$*True*, $D_{ij}\}$
9:     **else if** ($D_{ij} < \tau$) AND ($X_{ij} < \eta$) **then** return $\{$*True*, $D_{ij}\}$
10:    **else** return *False*

---

Since a graph obtained by such an analysis may be quite dense, we perform a *transitive reduction* operation on the graph to remove redundant edges.

## C  Extended Experiments/Baselines

This section presents a detailed breakdown of the quantitative results discussed in Section 4.3. We evaluate ReViND against four baselines in 15 environments with varying levels of complexity, in terms of environment organization, obstacles, and scale. Tables 2 and 3 summarize the performance of the different methods for the task of maximizing the $R_{\text{grass}}$ and $R_{\text{sun}}$ utilities, respectively.

We see that ReViND is able to consistently outperform the baselines, both in terms of success as well as its ability to maximize the utilities $\mathbb{1}$. In particular, we see that ReViND's performance closely matches that of IQL in the easier environments, where the system does not need to rely excessively on the graph. However, the real prominence of ReViND is evident in the more challenging environments, where it is consistency successful while also maximizing the chosen utility. As the horizon of the task increases, the search algorithm on the graph returns more desirable paths that may stray from the direct, shortest path to the goal, but are highly effective in maximizing the utility. We also note that ViNG, which uses a similar topological graph, is statistically similar to ReViND in terms of its goal-reaching ability; however, since it does not support a mechanism to customize the behavior of the learned policy, it suffers in the other performance metrics. BC, fBC and IQL consistently fail to reach goals beyond 15-20m away, due to challenges in learning a useful policy from offline data — these "flat" policies often demonstrate *bee-lining* behavior, driving straight to the goal, which leads to collisions in all but the easiest experiments.

## D  Formal Analysis of Proposition 3.1

**Proposition 3.1** *If we recover the optimal value function $V^*(s, s')$ for short-horizon goals $s'$ (relative to $s$), and $\mathcal{G} = \mathcal{S}$ (all states exist in the graph), and the MDP is deterministic with $\gamma = 1$, then finding the minimum-cost path in the graph $\mathcal{G}$ with edge-weights $-V^*(s, s')$ recovers the optimal path. .*

*Proof*: Let $A(s)$ and $A^h(s)$ define a set of all nodes adjacent to node $s$ and within a short horizon from a node $s$ correspondingly.

The Bellman equation can be used to write the cost of the minimal-cost path, $J^*(s, g)$, in the graph with rewards defined via edge-weights $r(s, a, s') = -V^*(s, s')$:

$$J^*(s, g) = \min_{s' \in A^h(s)} [-V^*(s, s') + J^*(s', g)] = -\max_{s' \in A^h(s)} [V^*(s, s') - J^*(s', g)].$$

We can expand the recursion:

$$J^*(s, g) = -\max_{s' \in A^h(s), s'' \in A^h(s'), \ldots, g \in A^h(s^{(n)})} [V^*(s, s') + V^*(s', s'') + \ldots + V^*(s^{(n)}, g)]. \quad (5)$$

We can further expand each $V^*(\cdot, \cdot)$ term as

$$V^*(s^{(n-k)}, s^{(n-k+1)}) = \max_{\substack{s_1 \in A(s^{(n-k)}) \\ s_2 \in A(s_1) \\ s^{(n-k+1)} \in A(s_t) \\ t \in \mathbb{N}}} [-C(s^{(n-k)}, s_1) - C(s_1, s_2) - \ldots - C(s_t, s^{(n-k+1)})].$$

$$(6)$$

If we expand every term in Equation 5 with 6 it becomes exactly the optimization objective for the shortest path problem with the original edge-weights. One can see $V^*(s, s')$ as a solution to the shortest path problems in the subgraphs of $\mathcal{G}$ induced by $A^h(s)$.

# E   Miscellaneous Implementation Details

Table 4 presents the neural network architectures used by our system. We provide the important hyperparameters for training our system in Table 5. The underlying learning algorithm in ReViND is based on IQL [3], and we encourage the reader to check out the IQL paper for more implementation details.

| Layer | Input Shape | Output Shape | Layer details |
|---|---|---|---|
| 1 | [3, 64, 48] | [1536] | Impala Encoder [47] |
| 2 | [1536] | [50] | Dense Layer |
| 3 | [50] | [50] | `tanh` (LayerNorm) |
| 4 | [50], [3] | [53] | Concat. image & goal |
| *Policy Network $a_t \sim \pi(s_t)$* | | | |
| 5 | [53] | [256] | Dense Layer |
| 6 | [256] | [256] | Dense Layer |
| 7 | [256] | [10] | Dense Layer |
| *Q Network $Q(s_t, a_t)$* | | | |
| 5 | [53], [10] | [256] | Dense Layer |
| 6 | [256] | [256] | Dense Layer |
| 7 | [256] | [1] | Dense Layer |
| *Value Network $V(s_t)$* | | | |
| 5 | [53] | [256] | Dense Layer |
| 6 | [256] | [256] | Dense Layer |
| 7 | [256] | [1] | Dense Layer |

**Table 4:** Architectures of the various neural networks used by ReViND.

# F   Interim Code Release

We are sharing the code corresponding to our offline learning algorithm, labeling, and evaluation scripts — please see `revind_code.zip` in the supplemental material. We will make a more polished and usable version of the code available with a future release.

| Hyperparameter | Value | Meaning |
|---|---|---|
| $\tau$ | 0.9 | IQL Expectile |
| A | 0.1 | Policy weight |
| $\gamma$ | 0.99 | Discount factor |
| $\eta$ | 0.005 | Soft Target Update |
| $\alpha_{\text{actor}}, \alpha_{\text{critic}}, \alpha_{\text{value}}$ | $3e-4$ | Learning rates |

**Table 5:** Hyperparameters used during training ReViND from offline data.

**Training Environments**

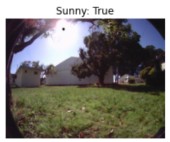 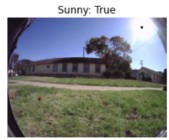 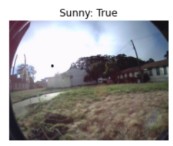 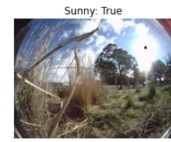 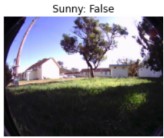

**Deployment Environments**

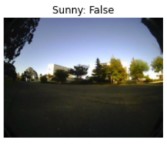 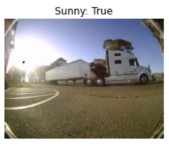 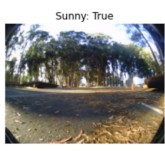 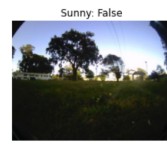 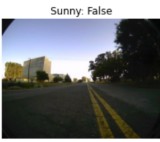

**Figure 6:** Example egocentric observations from the training dataset [5] (top) and the deployment environments (bottom), including the predicted labels for the "sunny" reward.

## G  Experiment Videos

We are sharing experiment videos of ReViND deployed on a Clearpath Jackal mobile robotic platform — please see `revind_video.mp4` in the supplemental material. The videos correspond to the qualitative behaviors learned by ReViND for the task of maximizing the corresponding utility functions, and a comparison to alternative learning-based algorithms (Table **??**).

## H  Environments

We train ReViND using 30 hours of publicly available robot trajectories collected using a randomized data collection procedure in an office park [5]. We conduct evaluation experiments in a variety of novel environments with similar visual structure and composition as the training environments — i.e. suburban environments with some traversals on the grass, around trees of a certain kind, and on roads. While it may be extremely challenging to get generalization capabilities that work for *all* scenarios, we demonstrate that ReViND can learn behaviors from a small offline dataset and generalize to a variety of previously unseen, *visually similar* environments including grasslands, forests and suburban neighborhoods. Figure 6 shows some example environments from the training and deployment environments, along with their corresponding labels (automatically generated).

