# OpenReview forum: "Offline Reinforcement Learning for Customizable Visual Navigation"
_NeurIPS.cc/2022/Workshop/Offline_RL — Offline RL Workshop NeurIPS 2022_

### Official Review · Reviewer_QvRH · 2022-10-18
**Interesting application of offline RL to robotic navigation; some clarifications/further ablations would be really helpful**

**Rating:** 6
**Confidence:** 4

**Review:**

Summary: This paper introduces ReViND, a robotic system that trains a navigational policy using generic, offline data and employs a self-supervised relabeling strategy to incorporate user-specified rewards. ReViND uses environment data to build a topological graph for navigation and uses the Q-function learned from the offline data via IQL to populate the edge weights. Then, it uses the Dijkstra algorithm to find the shortest path in the graph.  The works evaluate ReViND directly in the real-world robot and the experiments show improvements over prior baselines based on imitation learning, offline RL-only, and graph-based navigation-only.

Strengths:
- The work successfully applies offline RL methods in the real world without collecting environment-specific, online data for training.

- It also uses prior public datasets for training, making the method generic and agnostic to the robot or physical environment.

- It bridges different approaches (offline RL + topological graph search) and delivers gains over using them separately.

Major Concerns:
	My main concerns/questions are related to the fact that ReViND relies on human-teleoperated data collection to build the topological graphs. I believe the method is an interesting contribution, but I would recommend clarifying/addressing the following points in the paper:

- If the graph is built using previous data, the map would be static and it would not account for changes in the environment during the robot navigation (e.g., a person walking around). This seems an important limitation (and a possible safety concern) to be discussed in the paper.

- If the environment is explored and well-known prior to the robot navigation, which arguments justify the use of RL? Which priors are the RL method learning that ViNG does not have? Optimizing utility functions computed from the observation is not a sufficient argument since these could also be added directly as weights in the graph. In this line, this would be an interesting ablation to consider: ViNG + the utility functions in the graph edges (instead of a reward component).

- Given the human-teleoperated data, it would be possible to use hindsight goal relabeling and generate suboptimal demonstrations to train a BC-based method. That would be an important baseline to add and help clarify if the offline component is indeed necessary.


Minor concerns/suggestions:
- The font size of Figure 1 is too small. I would suggest making the figure bigger and increasing the font size to improve the illustration (the resolution seems OK).

- Appendix C cites Algorithm 0 (which does not exist), although it is linked to Algorithm 1.


I believe this work has potential impact and significance as a real-world robotic system deploying an offline-RL algorithm, which justifies its appearance as part of the workshop. There are some points to discuss in the paper, and I recommend the authors address them for the final version or future submissions.

---

### Official Review · Reviewer_mDyv · 2022-10-19

**Rating:** 6
**Confidence:** 3

**Review:**

The authors propose a combination of offline, goal-based Reinforcement Learning and topological planning for learning visual navigation of robots in the real world. Given some utility functions expressing human navigation preferences (e.g. staying in sunlight or in the grass), different reward functions are constructed and applied to the dataset. A topological graph approximating the connectivity of the world is constructed and used to search for subgoals that the agent will have to achieve. Experiments show that the method outperforms Behavioral Cloning algorithms and an RL algorithm that does not use a topological graph.

I found the paper not self-contained. It is difficult to understand the method used without having to look at the Appendix, while the main paper contains a lot of redundant parts. It is also not clear to me what is the interaction between goal-based planning, where the agent has to look for subgoals, and reward-based planning, where the agent has to optimize for different reward functions. I strongly suggest the authors include in the main paper details about the learning algorithm and how the connectivity graph is constructed.

I did not find the choice of the baselines well justified. In particular, I think that the topological graph is giving a strong bias and help in learning since it is built by manually driving the agent in the environment. Comparing this work to algorithms that do not use such information seems unfair. On the other hand, comparing algorithms that do not incorporate reward information on different reward functions also seems strange.

Nevertheless, I acknowledge that visual navigation using offline RL is a very challenging problem, and I think that this paper might be beneficial to discuss during the workshop.